# Effect of the Dry-Cured Fermented Sausage “Salchichón” Processing with a Selected *Lactobacillus sakei* in *Listeria monocytogenes* and Microbial Population

**DOI:** 10.3390/foods10040856

**Published:** 2021-04-15

**Authors:** Irene Martín, Alicia Rodríguez, Lourdes Sánchez-Montero, Patricia Padilla, Juan J. Córdoba

**Affiliations:** Food Hygiene and Safety, Meat and Meat Products Research Institute, Faculty of Veterinary Science, University of Extremadura, Avda. de las Ciencias. s/n, 10003 Cáceres, Spain; iremartint@unex.es (I.M.); aliciarj@unex.es (A.R.); lourdessv@unex.es (L.S.-M.); patriciapt@unex.es (P.P.)

**Keywords:** *L. monocytogenes* reduction, dry-cured fermented sausages, *L. sakei*, challenge test

## Abstract

In the present work, the effect of processing of dry-cured fermented sausage “salchichón” spiked with the selected *Lactobacillus sakei* 205 was challenge-tested with low and high levels of *L. monocytogenes*. The evolution of the natural microbial population throughout the “salchichón” ripening was also evaluated. For this, a total of 150 “salchichón” were elaborated and divided into six equal cases which were inoculated with different levels of *L. monocytogenes*, and *L. sakei* 205. Afterwards, sausages were ripened for 90 days according to a typical industrial process. Moisture content (%) and water activity (a_w_) decreased throughout the ripening up to values around 26% and 0.78, respectively. No differences for moisture content, a_w_, pH, NaCl and nitrite concentration were observed between the analyzed cases. Lactic acid bacteria counts in the *L. sakei* 205 inoculated cases were always higher than 6 log CFU g^−1^ during ripening. *Enterobacteriaceae* counts were reduced during ripening until non-detectable levels at the end of processing. Reductions in *L. monocytogenes* counts ranged from 1.6 to 2.2 log CFU g^−1^; therefore, the processing of “salchichón” itself did not allow the growth of this pathogen. Reduction in *L. monocytogenes* was significantly higher in the cases inoculated with *L. sakei* 205.

## 1. Introduction

Dry-cured fermented sausages are Mediterranean products that are widely consumed in Spain and well known in international markets [1,2]. Among the Spanish dry-cured fermented sausages, “salchichón” is a typical dry-cured fermented sausage classed as a ready-to-eat (RTE) food that is manufactured with traditional technologies without adding starter cultures [3]. This product is usually made from comminuted meat and fat, mixed with salt and other spices, and filled into casings, before their ripening in drying chambers, at temperature and relative humidity (RH) conditions that may vary between 7 and 14 °C, and 80 and 85%, respectively [4,5]. At these conditions, a decrease in water activity (a_w_) values below 0.90 and a slight reduction in pH to levels ranging from 5.4 to 6.0 have usually been reported for this kind of meat product [6,7].

The food-handling involved in the manufacture of “salchichón” increases the risk of microbial contamination, with *Listeria monocytogenes* being the most hazardous pathogenic microorganism in this RTE product. *L. monocytogenes* is the causative agent of listeriosis, one of the most serious foodborne diseases caused mainly by food consumption [8]. Dry-cured fermented sausages are considered products with low risk for foodborne listeriosis [9,10]; however, the presence of this pathogenic bacterium has been reported in ripened sausages [11,12,13], and in some cases, it has even been involved in listeriosis outbreaks [14,15].

In the European Union (EU), the food safety criteria for *L. monocytogenes* in RTE food products have been published in the Regulation (CE) 2073/2005 [16] amended by Regulation (CE) 1441/2007 [17]. This regulation set a maximum level of 100 CFU g^−1^ for *L. monocytogenes* in RTE foods throughout their shelf life. However, it has been found that dry-cured fermented sausages contaminated by *L. monocytogenes* at levels higher than 100 CFU g^−1^ are consumed without any further heat treatment in the EU. Thus, the development of *L. monocytogenes* in this RTE meat product should be considered a great public health concern [13].

In order to minimize the risk that the presence of *L. monocytogenes* implies in dry-cured fermented sausages, the implementation of a process addressing the survival of this pathogenic bacteria throughout the manufacturing procedure is of great importance [18]. Thus, reducing or eliminating *L. monocytogenes* from meat products is a real challenge for the meat industry and food safety authorities.

There are few references in the literature of a challenge test to evaluate the effect of processing of “salchichón” in the growth/inactivation of *L. monocytogenes*. Thus, it is not clear if *L. monocytogenes* can be diminished throughout the processing of this product. Different studies have indicated that this pathogen could survive during the dry-cured fermented sausage’s manufacturing and may not be completely eliminated [19], and in some cases, the growth of this microorganism was reported during ripening [20].

In the evaluation of the growth/inactivation of *L. monocytogenes* throughout the processing of “salchichón”, the effect of lactic acid bacteria (LAB) should be considered because selected LAB strains have demonstrated antimicrobial effect against this pathogenic bacterium [21,22]. Among LAB, *Lactobacillus sakei* has a technological use in the preservation of several dry-cured fermented sausages due to its capacity to produce organic acids, hydrogen peroxide, and bacteriocins [23]. In fact, *L. sakei* has shown protective antimicrobial effect against both pathogenic and spoiler microorganisms in meat products [24,25].

In addition to the evaluation growth/inactivation of *L. monocytogenes*, the effect of processing of “salchichón” on the evolution of a natural microbial population should be determined, because unusual development of some microbial groups may lead to irregular microbial quality of the sausages and even spoilage during ripening. For example, growth of some *Enterobacteriaceae* strains throughout ripening in dry-cured meat products has been reported [26], which may reduce the microbiological quality of the sausage pieces.

With the aim of evaluating potential food safety implications, “salchichón” spiked with selected strains of *L. sakei* was challenge-tested with low and high levels of *L. monocytogenes*. In addition, the effect of dry-cured fermented sausage “salchichón” processing on the evolution of the natural microbial population of this product was evaluated.

## 2. Materials and Methods

### 2.1. Microbial Cultures

The strain *L. sakei* 205 from the Food Hygiene and Safety Collection at the University of Extremadura was used for the inoculation of “salchichón”, with LAB as a protective culture. This strain was isolated from traditional dry-cured fermented sausages and selected by its antagonist activity against *L. monocytogenes* in agar “salchichón” (unpublished data). To prepare the inoculum of *L. sakei* 205, 100 μL of a stock culture (stored in brain heart infusion (BHI) broth (Conda, Spain) containing 20% (*w*/*v*) glycerol at −80 °C) were inoculated onto 10 mL of de Man–Rogosa–Sharpe (MRS) broth (Fisher Bioreagents, Belgium) and incubated for 48 h at 30 °C. At the end of the incubation, ≈8.0 log CFU mL^−1^ cells were obtained and an aliquot of this was diluted in 1% (*w*/*v*) peptone water (Conda, Spain) to reach a final concentration of approximately 6.0 log CFU mL^−1^. Then, the culture was centrifuged at 10,000× *g* for 5 min, and the supernatant was discarded. The sediment was then washed and resuspended in phosphate-buffered saline (PBS) and used for the inoculation of the “salchichón” mix before stuffing. To determine the final concentration (CFU mL^−1^) of *L. sakei* 205 in PBS in order to adjust the level of inoculation, serial dilutions in 1% (*w*/*v*) peptone water were inoculated onto MRS agar (Oxoid, UK) and incubated anaerobically at 30 °C for 72 h. In addition, initial levels of LAB on the sausages at day 0 of processing were determined as described previously.

For the inoculation of the “salchichón” with *L. monocytogenes*, strain S7-2 (serotype 4b) belonging to National Institute of Agricultural and Food Research and Technology (INIA) collection (Madrid, Spain) was used. To prepare the *L. monocytogenes* inoculum, 100 μL of a stock culture (stored in BHI broth containing 20% (*w*/*v*) glycerol at −80 °C) were transferred to 10 mL BHI broth and incubated for 24 h at 37 °C. A total of 100 μL of such culture were then transferred to a second tube of 10 mL BHI and incubated overnight at 37 °C. At the end of the incubation period, ≈8.0 log CFU mL^−1^ cells were obtained and aliquots of this were diluted to reach final concentrations of approximately 7.0 log CFU mL^−1^ and 4.0 log CFU mL^−1^. Then, the cultures were centrifuged at 10,000× *g* for 5 min, the supernatants discarded, and the sediments were washed and resuspended in PBS and used for the inoculation of the “salchichón” mix before stuffing. To verify the levels of inoculation, serial dilutions were inoculated onto Chromagar^TM^
*Listeria* agar plates and incubated at 37 °C for 48 h. In addition, the real initial levels (CFU g^−1^) of *L. monocytogenes* on the sausages were determined at day 0 of processing.

### 2.2. Preparation of Dry Fermented Sausages “Salchichón”

The mixture used for the manufacture of dry fermented sausages “salchichón” was purchased from a meat company in the Extremadura region (Cáceres) and its composition consisted of minced Iberian pork meat (90%) and Iberian pig fatback (7%), with an addition of NaCl (1.8%), cane sugar (0,4%), potassium nitrate (120 ppm), sodium nitrite (100 ppm), black paper and spices. This mixture was transported in refrigerated conditions (<2 °C) from the company to the meat pilot plant located at the Faculty of Veterinary of the University of Extremadura in order to prepare the sausages. Then, the mixture was divided into six equal cases of 10 kg each for the inoculation: (1) B (inoculated only with *L. sakei* at ≈6 log CFU g^−1^); (2) LI (inoculated with *L. monocytogenes* at ≈4 log CFU g^−1^); (3) LI+LAB (inoculated with *L. monocytogenes* at ≈4 log CFU g^−1^ combined with *L. sakei* at ≈6 log CFU g^−1^); (4) HI (inoculated with *L. monocytogenes* at ≈7 log CFU g^−1^); (5) HI+LAB (inoculated with *L. monocytogenes* at ≈7 log CFU g^−1^ combined with *L. sakei* at ≈6 log CFU g^−1^); and (6) C (uninoculated control).

In all cases, except in C, respective microorganism inocula were adjusted and prepared to be resuspended in a total volume of 150 mL of PBS (as described in Section 2.1) that were added to the ingredients and mixed with an automatic kneader, that was cleaned and sanitized between cases. In the case of C, 150 mL of sterilized PBS were added instead of the bacterium inocula.

The meat dough of each case was stuffed into regenerated collagen casings (40 mm in diameter) supplied by Viscofan (Navarra, Spain). The final weight of each sausage was approximately 500 g. The sausages obtained were ripened in controlled drying chambers at the Faculty of Veterinary Science of the University of Extremadura according to the industrial traditional conditions of “salchichón”: 5 °C at 85% relative humidity (RH) for 3 days, then 7 °C and 80% RH for the 17 days, 9 °C and 75% RH for 10 days, and finally, the sausages were kept at 12 °C and 70% RH to reach 90 days of ripening.

Five sausages of each case were taken at 0, 15, 30, 60, and 90 days of the ripening time for microbiological and physicochemical analysis. Before analysis, casings were aseptically removed in a laminar flow cabinet (Telstar, Spain). The experiment, consisting of 6 different cases, 5 sampling times, and 5 different analyzed sausages for each case and sampling time, was evaluated once, according to the European Union Reference Laboratory Technical Guidance Document for conducting shelf-life studies on *L. monocytogenes* in RTE foods (such as “salchichón”) where no growth or the growth probability of this pathogen is ≤10% [27].

### 2.3. Microbiological Analysis

#### 2.3.1. Confirmation of Absence on *L. monocytogenes* Contamination in Control

Control C was tested to confirm the absence of natural contamination of *L. monocytogenes*. For this, 25 g of each of the 5 sausages were taken at every sampling time and evaluated for the presence or absence of this pathogen according to ISO 11290-1 (International Organization for Standardization [28]).

#### 2.3.2. Estimation of Microbiological Levels

For the remaining microbiological analysis, 10 g of each of the 5 sausages were sampled at every sampling time, mixed with 90 mL of 1% (*w*/*v*) peptone water, and homogenized in a Stomacher machine (Seward, model 400 Circulator, West Sussex. UK) at 300 rpm for 1 min. Decimal serial dilutions were subsequently carried out in 1% (*w*/*v*) of peptone water, and then 100 µL of the cell suspensions were spread onto the surface of different agar plates according to the microbial group analyzed.

In all the inoculated cases with *L. monocytogenes* (LI, LI+LAB, HI, and HI+LAB), viable counts of this pathogen were enumerated on CHROMagar^TM^
*Listeria* Chromogenic media (CH-L, Scharlab, Barcelona, Spain) in duplicate. Each plate was seeded with 0.1 mL, incubated at 37 °C for 24 and 48 h. After the incubation period, the characteristic *L. monocytogenes* colonies, green colonies with a surrounded opaque halo, were counted.

In all cases, 5 groups of microorganisms were determined by using different culture media: the total viable microorganism counts on plate count agar (PCA; Conda Spain), LAB in MRS agar, *Enterobacteriaceae* on Violet Red Bile Glucose agar (VRBG, Conda, Spain), *Staphylococci* in Mannitol salt agar (MSA; Oxoid, UK) and mold and yeast counts on malt extract agar (MEA; 20 g/L of malt extract (Scharlab, Spain), 1 g/L of peptone water, 20 g/L of D (+) glucose monohydrate (Scharlab, Spain), bacto agar 20 g/L (Scharlab, Spain). All the above inoculated media were incubated at 30 °C for 48 h, except MEA and VRBG, which were cultured at 25 °C for 5 days and 37 °C for 24 h, respectively.

#### 2.3.3. Evaluation of Implantation of *L. sakei* 205

In addition, to evaluate the implantation of *L. sakei* 205, MRS plates at the last sampling time (90 days) of *L. sakei* 205 inoculated cases were taken, and 50% of the characteristic LAB colonies were randomly isolated and inoculated in MRS broth and streaked on fresh MRS agar plates and incubated at 37 °C. The cultures were sub-cultured to obtain pure cultures. Pure cultures were maintained in a sterilized MRS broth and kept at −20 °C until characterization by molecular analysis. This procedure was also followed to evaluate LAB colonies on MRS plates in case C at day 90 of ripening to confirm the absence of *L. sakei* 205. The identification of the LAB strains was performed by sequencing analysis of the 16S rRNA region according to the methodology proposed by Walter et al. [29], and PFGE analysis of the DNA with the restriction *NotI* and *SgsI* enzymes (Thermo Fisher Scientific, Waltham, MA., USA) following procedures previously described by Alía et al. [30].

### 2.4. Physicochemical Analysis

All the analyses for determination of the physicochemical characteristics of the sausages were made in quintuplicates in those cases in which *L. monocytogenes* was not inoculated (B and C).

#### 2.4.1. Water Activity Determination

The a_w_ of dry fermented sausages “salchichón” was determined at 25 °C by using a Novasina Lab Master Water activity meter model AW SPRINT-TH 300 (Novasina AG, Switzerland). Calibration was performed by using several saturated solutions of known a_w_.

#### 2.4.2. Moisture Content Determination

Moisture content was determined following the official methods of the Association of Official Analytical Chemists [31]. This parameter was determined gravimetrically.

#### 2.4.3. pH Determination

The pH was measured with a pH-meter (Model 340, Mettler-Toledo GmbH, Greifensee, Switzerland) that was calibrated with 3 different standard pH solutions (4.0, 7.0 and 9.25). The pH was determined after homogenizing 3 g of each sample with 27 mL of distilled water for 30 s using a homogenizer.

#### 2.4.4. Sodium Chloride Determination

NaCl was determined in duplicate for each of the 5 dry-cured sausages of each case at the end of the ripening period using the Volhard method [32].

#### 2.4.5. Nitrite Determination

Nitrite content was determined in each of the 5 dry-cured sausages of each case at the end of the ripening period according to the method described by AOAC (2005) [33]. Measurement of residual nitrite was spectrophotometrically conducted from pinkish dye produced by coupling sulfanilamide with NED dihydrochloride. A calibration curve was obtained by diluting the standard solution (100 mg/l NaNO_2_) with distilled water to cover a concentration range from 0.1 to 0.8 mg/l NaNO_2_. The residual nitrite content was calculated using a standard curve of nitrite solution as mg nitrite per kg sample.

### 2.5. Statistical Analyses

The statistical treatment was carried out using the software IBM SPSS Statistic version 20 (IBM, New York, NY, USA). For the statistical analysis of the data, different cases and days of ripening were used as independent variables. The counts (Log CFU g^−1^), a_w_, pH, moisture content, sodium chloride, and nitrite values were analyzed as dependent variables. Once the dependent and independent variables of the analysis were determined, a study of the normality of the different data populations was carried out using the Shapiro–Wilk test. Subsequently, the analysis of the data was conducted using the Mann–Whitney test [34]. Statistical significance was established at *p* ≤ 0.05.

## 3. Results

### 3.1. Enumeration of Microorganisms

Microbiological analysis of the dry-cured sausages to monitor the dynamic changes in the populations responsible for the ripening was carried out. The results obtained from the enumeration of microorganisms of the six cases are shown in Table 1. Total aerobic microorganism counts showed average levels higher in the cases inoculated with *L. sakei* 205 (B, LI+LAB, HI+LAB) than in the uninoculated control (C) or in the cases only inoculated with *L. monocytogenes* (LI, HI). There was not an increase in the levels of the total aerobic microorganisms throughout the ripening of “salchichón” in any of the analyzed cases.

Initial *Enterobacteriaceae* counts ranged between 4.09 and 3.82 log CFU g^−1^ in all analyzed cases (Table 1). However, the number of *Enterobacteriaceae* always significantly decreased (*p* ≤ 0.05) throughout the ripening and were below the detection limit at the end of the maturation process.

Counts on MSA agar (*Staphylococci*) showed initial levels of ≈ 5 log CFU g^−1^ in C and B (Table 1). In both cases, an increase (*p* ≤ 0.05) in this microbial group of about 1 log CFU g^−1^ was observed until day 30 of ripening. In C, such counts were kept constant until the end of the processing; however, in B, the levels of *Staphylococci* decreased until initial levels at the two last sampling days. Although in the cases where *L. monocytogenes* + *L. sakei* 205 were inoculated there were no data at days 0 and 15 of analysis, the evolution of *Staphylococci* at days 30, 60 and 90 was similar to that the observed in case B, showing levels at the end of the ripening period of at least 1 log CFU/g lower (*p* ≤ 0.05) than in the uninoculated control C.

Counts on MEA agar (molds and yeasts) ranged between 5.3 and 7.1 log CFU/g during the ripening process in all the analyzed cases (Table 1). No significant effect of the ripening on levels of molds and yeasts was observed.

The LAB levels are shown in Figure 1. The microbiological analysis revealed significant differences at the beginning of the ripening between cases inoculated with LAB (B, LI+LAB, HI+LAB) and those which were not inoculated with LAB (C, LI, HI). Counts on MRS agar were around 7 log CFU g^−1^ for all cases inoculated with *L. sakei* (B, LI+LAB, HI+LAB) and 4–5 log CFU g^−1^ for those not inoculated with *L. sakei* (C, LI, HI). LAB counts of the cases inoculated with *L. sakei* remained constant (≈7 log CFU g^−1^) until 30 days of ripening, then they decreased slightly but their levels were always higher than 6 log CFU g^−1^. The levels of LAB in those cases that were not inoculated with *L. sakei* showed an increase in these counts at days 15 and 30 of ripening, reaching counts around 7 log CFU g^−1^ (Figure 1). Thereafter, counts decreased throughout the ripening process, except in C, where the counts on MRS agar were more similar to those found in the cases inoculated with *L. sakei* 205. However, the cases inoculated only with *L. monocytogenes* (LI and HI) showed lower LAB counts throughout the maturation process, and were significantly different from the remaining cases at the end of the ripening.

When isolates of the cases inoculated with *L. sakei* were analyzed to evaluate the implantation of this strain, most of the investigated isolates (85.7%) were identified as *L. sakei* (100% identity) by sequencing analysis of the 16S rRNA region. In addition, these isolates showed the same pattern of *L. sakei* 205 in the PFGE analysis. The remaining strains were *Lactobacillus plantarum* group (7.15%) and *Lactobacillus curvatus* (7.15%). However, the isolates of C, LI and HI were identified as *Lactobacillus curvatus* (50%), *Enterococcus faecalis* (36%), and *Lactobacillus plantarum* groups (14%).

The inoculation of *L. monocytogenes* resulted in an initial concentration of 6.63–6.60 log CFU g^−1^ in HI and HI+LAB and 4.09–4.18 CFU g^−1^ in LI and LI+LAB (Table 2). *L. monocytogenes* did not grow in any of the inoculated cases at any of the ripening time. Levels of this pathogenic bacterium showed significant (*p* ≤ 0.05) decreases in comparison with the initial ones, at day 15 (LI+BAL) and at days 30, 60 and 90 (all cases inoculated with *L. monocytogenes*; Table 2). After 90 days of ripening, the reduction in *L. monocytogenes* counts in LI was 1.61 log CFU g^−1^, while in LI+LAB it was significantly higher (2.03 log CFU g^−1^, Figure 2). In HI, the reduction was 1.64 log CFU g^−1^, while in HI+LAB it was significantly higher (1.77 log CFU g^−1^, Figure 2).

### 3.2. Physicochemical Parameters

The evolution of moisture content (%), a_w_, and pH throughout the ripening process of cases C and B of “salchichón” is shown in Table 3. Initial moisture content in both cases was above 85%, decreasing throughout the ripening process until values of 25–26% (*p* ≤ 0.05; Table 3). However, there were no significant differences (*p* > 0.05) between both cases at any of the ripening times. The a_w_ decreased (*p* ≤ 0.05) from initial values (0.946–0.947) found in the raw product to values below 0.790 at day 90 (Table 3). No significant differences (*p* > 0.05) among control and inoculated cases were observed in this parameter.

The evolution of pH differed between cases C and B, showing lower values in the case inoculated with *L. sakei* than in the uninoculated control at days 15 and 30 of ripening (Table 3). However, at day 30, an increase in the pH value was observed in C up to values close to 5.9, and they were kept constant until the end of ripening. There were no significant differences between both cases at days 60 and 90 of processing.

NaCl and nitrite contents were only determined in ripened “salchichón” (at day 90), showing both cases (C and B) with similar values, higher than 3.2% NaCl and around 6.9–7.4 ppm of nitrite (Table 4). No differences (*p* > 0.05) between both cases in these parameters were found.

## 4. Discussion

In this work, the survival and control of *L. monocytogenes* throughout the processing of dry-cured fermented sausage “salchichón” was evaluated by using the challenge test [27]. For this study, an industrial traditional procedure for the production of “salchichón” was carried out [35]. Differences in the reduction/inactivation of *L. monocytogenes* in dry-cured fermented sausages in relation to the initial load have been reported [19,36]; therefore, two levels of inoculation of *L. monocytogenes* were used for this study: high level (about 7 log CFU g^−1^) and low level (around 4 log CFU g^−1^). In addition, exploring greater effectiveness in reducing *L. monocytogenes*, a selected strain of *L. sakei* (205) isolated from dry-cured fermented sausages and previously selected because of its antagonist activity against *L. monocytogenes* (unpublished data), was evaluated in the challenge test.

Physicochemical evaluation of case B inoculated with *L. sakei* 205 and uninoculated control (C) showed that the ripening process was correctly conducted. Thus, moisture content and a_w_ decreased throughout the ripening process, from initial values around 85% to 25–26% and from 0.94 to 0.78 a_w_, respectively, reaching values similar to those usually reported for dry-cured fermented sausages [37,38,39]. The decrease in a_w_ in dry-cured fermented sausages such as “salchichón” is important for extending the shelf life and safety of the product [40]. No differences for moisture content or a_w_ parameters were observed between both cases. This evidences that *L. sakei* 205 does not provoke any modification in any of the two above-mentioned parameters, as has been reported for other LAB assayed [41,42]. However, the strain *L. sakei* 205 provoked a slight reduction in pH in the first 30 days of ripening, in comparison with uninoculated control, probably due to the increase in the lactic acid content, as a result of carbohydrate breakdown by microbial metabolism [43]. There was no reduction in pH in the uninoculated control (C), in spite of LAB counts from natural contamination reaching similar levels as those found in B (inoculated with *L. sakei* 205). In the last stages of processing, an increase in pH was detected, mainly in B (inoculated with *L. sakei* 205), which could be explained by the accumulation of non-protein nitrogen and amino acid catabolism products [43,44]. Consequently, no differences between B and C were detected for this parameter at day 90 of ripening. There were also no differences between both cases (B and C) in the remaining physical–chemical parameters analyzed, NaCl and nitrite content. Levels of NaCl found at day 90 were similar to those reported for dry-cured fermented sausages [45] and in accordance with the level of salt used in the manufacture of sausages. Regarding nitrite content, low levels (lower than 10 ppm) were detected at the end ripening. Nitrite values obtained were in levels usually found in ripened sausages, because it is a very reactive compound and only residual nitrite that has not reacted with myoglobin is detected in finished products [46].

Besides the physicochemical parameter, the evolution of the different microbial groups in the six tested cases throughout the processing of “salchichón” was evaluated.

Levels of total aerobic microorganisms were higher in sausages inoculated with *L. sakei* 205 than in the remaining cases at the beginning of ripening, due to the effect of the addition of the protective cultures [22]. However, during processing there was not an increase in this microbiological parameter in any of the analyzed cases, mainly due to the decrease in the a_w_ until values around 0.78 at the end of maturation but also by the slight decrease in pH, and the presence of NaCl and nitrites. These data are consistent with those observed in other dry-cured fermented sausages [46,47,48].

LAB counts of the cases inoculated with *L. sakei* 205 remained constant, with values of about 7 log CFU g^−1^ until 30 days of ripening and then a slight decrease was observed, probably due to the reduction in a_w_, but counts were always higher than 6 log CFU g^−1^. In addition, the characterization of the LAB isolates in these sausages showed that *L. sakei* 205 was the strain mostly detected, which means that this strain was properly implanted throughout the ripening. The levels of LAB in cases that were not inoculated with *L. sakei* 205 showed an increase in this microbial group at the beginning of processing to reach similar counts to those detected in the inoculated cases with the selected strain of LAB. However, in the analysis of the isolates no *L. sakei* was detected, which suggests that this strain only was present in cases inoculated with this selected LAB.

During the evolution of the remaining microbial groups, the decrease in *Enterobacteriaceae* counts until non-detectable levels at day 90 of ripening in all analyzed cases and the decrease in *Staphylococci* only in cases inoculated with *L. sakei* 205 at days 60 and 90 of ripening is relevant. The *Enterobacteriaceae* results are consistent with those reported by Cocolin et al. [49], who demonstrated the persistence of this microbial group until day 60 of ripening of dry-fermented sausages. It should be emphasized that no growth of *Enterobacteriaceae* during processing was found in this work; however, punctual growth of this microbial group during the processing of dry-cured fermented sausages has previously been reported [46,50]. The decrease in a_w_, the various metabolites excreted by LAB, and the slight drop in pH may partially explain the reduction and disappearance of *Enterobacteriaceae* in this kind of meat products [46]. The reduction in levels of *Staphylococci* only in cases inoculated with *L. sakei* 205 at the end of ripening may be related to the effect of the synergist action of this strain together with the reduction in a_w_ throughout the processing.

The evaluation of the growth/inactivation rate of *L. monocytogenes* in both high and low inoculation levels showed no growth of this pathogen at any of the ripening times evaluated. This aspect is relevant because at the first 15 days of ripening there could be conditions of temperature (7 °C), a_w_ (0.947), and pH (5.8–5.4) which can favor the growth of *L. monocytogenes*, but the synergistic effect of the above parameters and the presence of NaCl, nitrite, and LAB inoculated or from natural contamination inhibit the growth of this pathogen.

Throughout the ripening process, reductions in *L. monocytogenes* ranging from 1.6 to 2.2 log CFU g^−1^ were observed at both high and low inoculation levels of “salchichón”. These values are lower than reductions of up to 5 log CFU g^−1^ reported in inoculated Portuguese “linguiça” smoked dry-cured sausages [51], although in that work the reduction was mainly due to the smoking and the high temperatures employed in the process. However, reduction levels found in the present work are similar to those found for this pathogen in other dry-cured meat products [52]. Several authors have reported that a long ripening period, which is related to a high decrease in a_w_, leads to a higher reduction in *L. monocytogenes* counts in dry-cured meat products [53,54,55]. However, short ripening periods in dry-cured fermented sausages have been associated with a greater survival of *L. monocytogenes* [19,45]. Nitrite used in the formulation of “salchichón” at a concentration lower than 150 ppm, as occurred in the present work, also contributes to the reduction in *L. monocytogenes*, as has been reported in different types of dry-fermented sausages [56,57].

There were no differences in reductions in relation to the level of inoculation of *L. monocytogenes*. However, the reduction in the pathogen counts was significantly higher in HI+LAB and LI+LAB than in HI and LI. This additional reduction provoked by the presence of *L. sakei* 205 was 0.42 log CFU g^−1^ in LI. This proves that *L. sakei* 205 has anti-*L. monocytogenes* activity during the ripening of “salchichón”, even considering that in the cases inoculated only with *L. monocytogenes*, indigenous LAB were found, probably with some antimicrobial effect. *L. sakei* is highly adapted to the fermented meat matrix [39], and many studies have determined that *L. sakei* has been widely used as a biocontrol for *L. monocytogenes* in dry fermented sausages [24,58].

Although the additional reduction in *L. monocytogenes* provoked by the selected *L. sakei* was not very high, it could be sufficient to guarantee the elimination of this pathogenic bacterium throughout the processing of “salchichón” when this pathogen contaminates this product at the usual levels (below 2 log CFU g^−1^). This is very important, because minimizing the risk of listeriosis caused by the consumption of “salchichón” improves food safety and meets the microbiological criteria of RTE foods throughout their shelf life in the EU [1,2].

## 5. Conclusions

The processing of “salchichón” does not allow the growth of *L. monocytogenes.* On the contrary, it provokes a reduction in this pathogen that could be even higher by using the strain *L. sakei* 205 as a protective culture. The manufacturing of “salchichón” also allows minimization of *Enterobacteriaceae* until non-detectable levels. These findings may be of great interest both for the safety and extending the shelf-life of “salchichón”.

## Figures and Tables

**Figure 1 foods-10-00856-f001:**
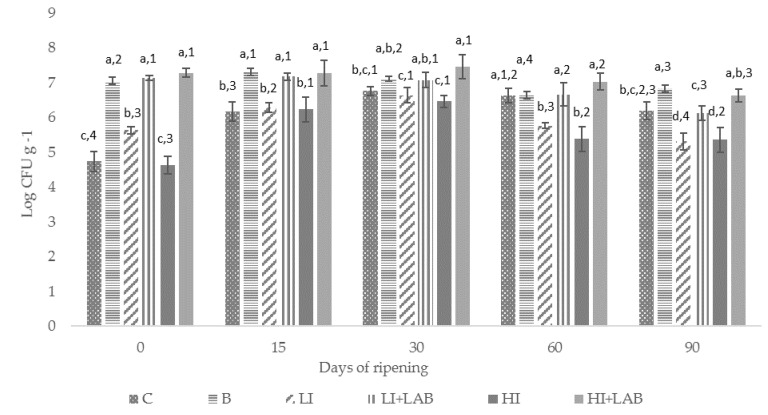
Evolution of lactic acid bacteria counts throughout the ripening process of “salchichón”. C (uninoculated control), B (inoculated with *L. sakei*), LI+LAB (inoculated with *L. monocytogenes* at ≈4 log CFU g^−1^ combined with *L. sakei*), HI+LAB (inoculated with *L. monocytogenes* at ≈7 log CFU g^−1^ combined with *L. sakei*), LI (inoculated with *L. monocytogenes* at ≈4 log CFU g^−1^), HI (inoculated with *L. monocytogenes* at ≈7 log CFU g^−1^). Bars with different letters (a–d) indicate significant differences (*p* ≤ 0.05) between cases on the same day. Bars with different numbers (1–4) indicate significant differences (*p* ≤ 0.05) between days in the same case.

**Figure 2 foods-10-00856-f002:**
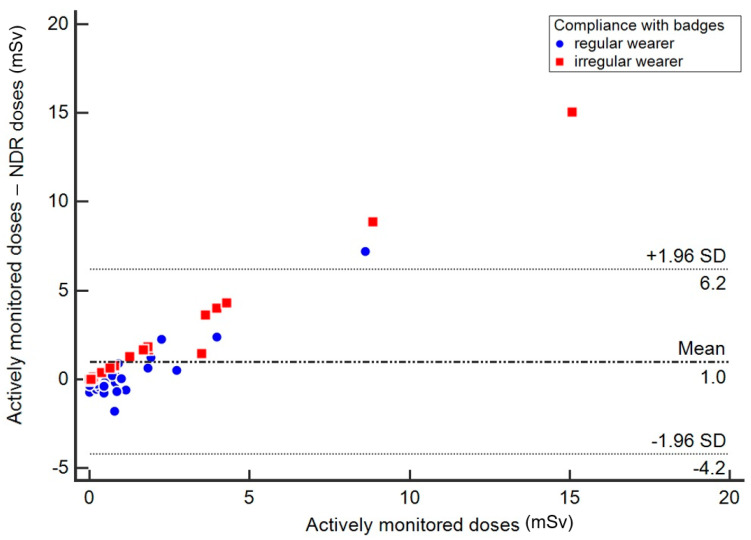
Reduction in the *L. monocytogenes* levels throughout the processing of dry-cured fermented “salchichón”. LI (inoculated with *L. monocytogenes* at ≈4 log CFU g^−1^), LI+LAB (inoculated with *L. monocytogenes* at ≈4 log CFU g^−1^ combined with *L. sakei*), HI (inoculated with *L. monocytogenes* at ≈7 log CFU g^−1^), HI+LAB (inoculated with *L. monocytogenes* at ≈7 log CFU g^−1^ combined with *L. sakei*). Bars with different letters (a,b) indicate significant differences (*p* ≤ 0.05) between cases on the same day.

**Table 1 foods-10-00856-t001:** Evolution of the different microbial groups in the different cases of dry-cured fermented sausages “salchichón” throughout the ripening process. Microorganisms determined in each media were: Total aerobic counts on plate count agar (PCA), *Enterobacteriaceae* on Violet Red Bile Glucose agar (VRBG), *Staphylococci* on Mannitol salt agar (MSA), and molds and yeasts on Malt extract agar (MEA).

		Days of Ripening				
Cases		0	15	30	60	90
C	PCA	5.45 ± 0.209 ^c,2^	6.56 ± 0.159 ^a,1^	6.86 ± 0.101 ^a,1^	6.62 ± 0.266 ^b,1^	6.68 ± 0.306 ^a,b,1^
	VRBG	4.01 ± 0.153 ^a,1^	3.48 ± 0.360 ^d,2^	2.11 ± 0.312 ^3^	1.99 ± 0.007 ^a,3^	nd
	MSA	5.04 ± 0.226 ^2^	5.94 ± 0.217 ^b,1^	6.08 ± 0.077 ^b,1^	6.02 ± 0.161 ^a,1^	5.90 ± 0.185 ^a,1^
	MEA	5.32 ± 0.281 ^b,3^	6.38 ± 0.188 ^b,1,2^	6.68 ± 0.074 ^a,b,1^	6.66 ± 0.158 ^a,1^	6.24 ± 0.100 ^b,2^
B	PCA	6.98 ± 0.135 ^b,1^	6.43 ± 0.328 ^a,2^	6.43 ± 0.328 ^b,2^	6.89 ± 0.170 ^a,b,1^	7.10 ± 0.118 ^a,1^
	VRBG	3.85 ± 0.109 ^a,b,1^	3.54 ± 0.130 ^c,d,2^	2.11 ± 0.271 ^3^	1.97 ± 0.012 ^a,b,3^	nd
	MSA	4.59 ± 0.428 ^2^	6.33 ± 0.074 ^a,1^	6.33 ± 0.074 ^a,1^	5.08 ± 0.397 ^b,2^	5.05 ± 0.208 ^b,2^
	MEA	7.16 ± 0.083 ^a,1^	7.04 ± 0.151 ^a,1,2^	6.92 ± 0.097 ^a,1,2,3^	6.70 ± 0.209 ^a,3^	6.81 ± 0.092 ^a,2,3^
LI+LAB	PCA	7.10 ± 0.049 ^a,b,1^	6.20 ± 0.068 ^a,3^	6.20 ± 0.068 ^b,3^	6.75 ± 0.281 ^b,2^	6.54 ± 0.199 ^b,2^
	VRBG	3.82 ± 0.057 ^b,1^	3.91 ± 0.047 ^b,1^	2.07 ± 0.211 ^2^	1.97 ± 0.007 ^a,b,2^	nd
	MSA	-	-	5.26 ± 0.320 ^e,1^	4.99 ± 0.243 ^b,1^	4.50 ± 0.225 ^c,2^
	MEA	-	-	6.46 ± 0.313 ^b,c,1^	6.27 ± 0.205 ^b,1,2^	5.92 ± 0.102 ^c,3^
HI+LAB	PCA	7.36 ± 0.212 ^a,1^	6.46 ± 0.084 ^a,2^	6.46 ± 0.084 ^b,2^	7.36 ± 0.372 ^a,1^	6.87 ± 0.237 ^a,b,2^
	VRBG	4.09 ± 0.215 ^a,1^	3.83 ± 0.069 ^b,c,2^	2.05 ± 0.132 ^3^	1.98 ± 0.014 ^a,b,3^	nd
	MSA	-	-	5.60 ± 0.024 ^d,1^	4.20 ± 0.117 ^c,3^	4.84 ± 0.172 ^b,c,2^
	MEA	-	-	6.73 ± 0.093 ^a,b,1^	6.66 ± 0.091 ^a,1^	6.06 ± 0.155 ^b,c,2^
LI	PCA	5.19 ± 0.251 ^c,3^	5.29 ± 0.327 ^c,2,3^	5.82 ± 0.245 ^c,1^	5.72 ± 0.256 ^c,1,2^	5.81 ± 0.271 ^c,1^
	VRBG	3.84 ± 0.246 ^a,b,2^	4.28 ± 0.139 ^a,1^	<1.97 ± 0.018 ^3^	<1.95 ± 0.033 ^b,3^	nd
	MSA	-	-	5.42 ± 1.120 ^e,1^	3.66 ± 0.335 ^c,3^	4.83 ± 0.094 ^b,c,2^
	MEA	-	-	5.38 ± 0.267 ^d,2^	6.05 ± 0.163 ^b,1^	5.98 ± 0.278 ^b,c,1^
HI	PCA	6.47 ± 0.010 ^b,1^	5.75 ± 0.118 ^b,3^	5.44 ± 0.130 ^d,4^	6.10 ± 0.089 ^c,2^	5.95 ± 0.219 ^c,2,3^
	VRBG	4.05 ± 0.131 ^a,1^	3.98 ± 0.167 ^b,2^	<2.18 ± 0.212 ^3^	<1.98 ± 0.005 ^a,b,3^	nd
	MSA	-	-	5.84 ± 0.157 ^c,1^	3.76 ± 0.362 ^c,3^	5.06 ± 0.115 ^b,2^
	MEA	-	-	6.09 ± 0.193 ^c^	6.02 ± 0.135 ^b^	6.04 ± 0.077 ^b,c^

C (uninoculated control), B (inoculated with *L. sakei*), LI+LAB (inoculated with *L. monocytogenes* at ≈4 log CFU g^−1^ combined with *L. sakei*), HI+LAB (inoculated with *L. monocytogenes* at ≈7 log CFU g^−1^ combined with *L. sakei*), LI (inoculated with *L. monocytogenes* at ≈4 log CFU g^−1^), HI (inoculated with *L. monocytogenes* at ≈7 log CFU g^−1^). Values are expressed as mean ± standard deviation. The means with different letters (a–e) in the same column indicate significant differences (*p* ≤ 0.05) between cases on the same day. Mean values with different numbers (1–4) in the same row indicate significant differences (*p* ≤ 0.05) between days in the same case and the same culture medium. nd: not detected (below the detection limit). (-): not determined.

**Table 2 foods-10-00856-t002:** Evolution of *L. monocytogenes* counts on inoculated cases of dry-cured fermented sausages “salchichón” throughout the ripening process.

Cases	Days of Ripening				
	0	15	30	60	90
LI	4.09 ± 0.091 ^1^	3.68 ± 0.451 ^1,2^	3.45 ± 0.233 ^2^	2.89 ± 0.136 ^3^	2.49 ± 0.231 ^b,3^
LI+LAB	4.18 ± 0.089 ^1^	3.84 ± 0.103 ^2^	3.42 ± 0.215 ^3^	2.74 ± 0.117 ^4^	2.14 ± 0.127 ^a,5^
HI	6.63 ± 0.056 ^1^	6.43 ± 0.179 ^1,2^	6.31 ± 0.075 ^b,2^	5.12 ± 0.214 ^3^	5.00 ± 0.104 ^b,3^
HI+LAB	6.60 ± 0.107 ^1^	6.43 ± 0.136 ^1^	6.04 ± 0.129 ^a,2^	4.99 ± 0.111 ^3^	4.83 ± 0.091 ^a,3^

LI+LAB (inoculated with *L. monocytogenes* at ≈4 log CFU g^−1^ combined with *L. sakei*), HI+LAB (inoculated with *L. monocytogenes* at ≈7 log CFU g^−1^ combined with *L. sakei*), LI (inoculated with *L. monocytogenes* at ≈4 log CFU g^−1^), HI (inoculated with *L. monocytogenes* at ≈7 log CFU g^−1^). Values are expressed as mean ± standard deviation. Mean values with different letters (a,b) in the same column indicate significant differences (*p* ≤ 0.05) between cases on the same incubation day. Mean values with different numbers (1–4) in the same row indicates significant differences (*p* ≤ 0.05) between incubation days in the same case.

**Table 3 foods-10-00856-t003:** Moisture content, water activity (a_w_) and pH in uninoculated control (C) and case inoculated only with *L. sakei* 205 (case B) of dry-cured fermented sausages “salchichón” throughout the ripening process.

	Cases	Days of Ripening				
		0	15	30	60	90
Moisture content (%)	C	89.67 ± 0.656 ^1^	56.35 ± 2.525 ^2^	43.16 ± 1.396 ^3^	32.34 ± 2.155 ^3,4^	26.04 ± 1.484 ^4^
	B	85.84 ± 0.235 ^1^	61.19 ± 3.052 ^2^	42.70 ± 1.387 ^3^	31.65 ± 1.188 ^4^	25.17 ± 1.484 ^5^
a_w_	C	0.947 ± 0.001 ^1^	0.905 ± 0.009 ^2^	0.874 ± 0.003 ^3^	0.821 ± 0.003 ^4^	0.785 ± 0.008 ^5^
	B	0.946 ± 0001 ^1^	0.914 ± 0.007 ^2^	0.877 ± 0.005 ^3^	0.817 ± 0.003 ^4^	0.779 ± 0.010 ^5^
pH	C	5.83 ± 0.226 ^1,2^	5.76 ± 0.058 ^a,2^	5.89 ± 0.030 ^a,1,2^	5.99 ± 0.071 ^1^	5.87 ± 0.105 ^1,2^
	B	5.82 ± 0.021 ^1,2^	5.48 ± 0.058 ^3^	5.73 ± 0.085 ^2^	5.98 ± 0.050 ^1^	5.91 ± 0.041 ^1^

Values are expressed as mean ± standard deviation. Mean values with different letters (a) in the same column indicate significant differences (*p* ≤ 0.05) between cases on the same day. Mean values with different numbers (1–4) in the same row indicate significant differences (*p* ≤ 0.05) between incubation days in the same case.

**Table 4 foods-10-00856-t004:** Sodium chloride (NaCl) and nitrite contents in uninoculated control (C) and case inoculated only with *L. sakei* 205 (B) in dry-cured fermented sausages “salchichón”.

Cases	NaCl (%)	Nitrites (ppm)
C	3.29 ^1^ ± 0.100	7.49 ± 1.019
B	3.36 ± 0.099	6.96 ± 0.770

^1^ Values are expressed as mean ± standard deviation. Mean values with different letters in the same column indicate significant differences (*p* ≤ 0.05) between cases.

## Data Availability

Not applicable.

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
