# Peer review of "Effect of the Dry-Cured Fermented Sausage “Salchichón” Processing with a Selected Lactobacillus sakei in Listeria monocytogenes and Microbial Population"

_foods, 2021, doi:10.3390/foods10040856_

Round 1

Reviewer 1 Report

The manuscript has been significantly improved as request

Kind regards

Author Response

We thank Reviewer 1 for the revision of the manuscript and the Review Report Form. There are not comments or suggestion in this second revision.  

Reviewer 2 Report

In my opinion the Authors addressed all the criticisms raised up by reviewers.

Author Response

We thank Reviewer 2 for the revision of the manuscript and the Review Report Form. There are not comments or suggestion in this second revision. 

Reviewer 3 Report

The current article has been improved.

However, some points should be clarified to be better understood by the readers.

The product was manufactured using one batch of meat, so replace Batches A, B etc. with CASES A, B etc.

L298-299 L. plantarum cannot be differentiated from L. paraplantarum or L. pentosus using 16srRNA gene sequence. Authors need to replace L plantarum with L plantarum group, if no specific PCR was applied. See Torriani et al., 2001

Author Response

We thank Reviewer 3 for the revision of the manuscript and the Review Report Form.

We have prepared a revised version of the manuscript considering the two requirements of Reviewer 3 as follows:

Comment 1: The product was manufactured using one batch of meat, so replace Batches A, B etc. with CASES A, B etc.

Response: Batches A, B, have been replaced by cases throughout the manuscript: in abstract lines 11, 15, 16 and 19, in material and methods lines from 119 to 215, results lines 226 to 341 and discussion lines 353 to 439.

Comment 2: L. plantarum cannot be differentiated from L. paraplantarum or L. pentosus using 16srRNA gene sequence. Authors need to replace L plantarum with L plantarum group.

Response: L. plantarum have been replaced by L. plantarum group in the revised manuscript (lines 282 and 284).

All changes have been clearly highlighted in the manuscript by using the "Track Changes" function in Microsoft Word.

This manuscript is a resubmission of an earlier submission. The following is a list of the peer review reports and author responses from that submission.

Round 1

Reviewer 1 Report

Manuscript ID: foods-1145772

Effect of the dry-cured fermented sausage “salchichón” processing with a selected Lactobacillus sakei on L. monocytogenes and microbial population during ripening

I think it's a very interesting and very important topic for food hygiene and technology nowadays. The contamination of L. monocytogenes of RTE foods and challenge test are object of studies in the food safety field. The topic is of interest for the academics and for the industry because of the results obtained and because of the novelty of the application in field. There are some studies like this in literature, but not specific in this kind of RTE product.

The manuscript evaluate the presence and contamination of “salchichon” by L. monocytogenes during ripening after inoculation of starter LAB; the research is well performed, the sampling and analysis were well done.

The manuscript is well written and easy to understand by readers. I believe that this manuscript does not need big changes.

  • LINE 27: …well known in international markets.. please add references
  • LINE 90: how do you evaluate the final concentration?

Statistical analysis was well performed

Results and Discussion were well explained

Reviewer 2 Report

The manuscript deals with the effect of processing of dry-cured fermented sausage “salchichón”. Trials with a selected Lactobacillus sakei (205) were performed and the effect against  L. monocytogenes was assessed. The manuscript is interesting and well build up. Methods are clearly described and generated data are robust. Results are well discussed.

In my opinion an English language revision is needed for a number a sentences. 

Please, revise/reduce the title; it is not clear the terme "on" before L. monoc.

Line 140: please, use italic for L. monocytogenes. please, also check all the text for the use of italic writing microbial species.

Reviewer 3 Report

The current article has focused on the processing of dry-cured fermented sausage “salchichón” with the addition of a protective culture (Lactobacillus sakei 205) and, co-inoculated with Listeria monocytogenes in 2 inoculum levels with or without the use of the protective culture. At the product, several analyses were performed (moisture content, aw, pH, NaCl and nitrite concentration) along with the microbial analysis and the molecular analysis. In general, the paper is providing some insights for the risk assessment, regarding the production of the sausage. However, the MS deals with retrospective methods providing basic research and the topic is not studied in depth.   

A few of the major problems in the study are the following:

At materials section:

There are not enough independent experiments performed in this study, only one batch for each case cannot provide sufficient results which are not biased.

L167-169: How many isolates were retrieved for the molecular identification.

At results section:

Table 1. how the authors explain the absence of Staphylococci and Yeasts and molds at 0 and 15 days of ripening and the high diversity between batches during the whole ripening period?

L267-270: insufficient, provide more details.

L279-289/ Table 2: a difference about 0.2 log CFU/g between control samples and samples inoculated with L. sakei can be significant?

L449-450: the differences in the counts are in between the deviation limits of the enumeration method.

Discussion section is very poor and repetitive.

In total, at its present form, the MS does not cover knowledge gaps and does not provide novel information.